# Functionalisation of Inorganic Material Surfaces with Staphylococcus Protein A: A Molecular Dynamics Study

**DOI:** 10.3390/ijms23094832

**Published:** 2022-04-27

**Authors:** Mohammed A. H. Farouq, Karina Kubiak-Ossowska, Mohammed M. Al Qaraghuli, Valerie A. Ferro, Paul A. Mulheran

**Affiliations:** 1Department of Chemical and Process Engineering, University of Strathclyde, 75 Montrose Street, Glasgow G1 1XJ, UK; mohammed.al-qaraghuli@strath.ac.uk (M.M.A.Q.); paul.mulheran@strath.ac.uk (P.A.M.); 2Department of Physics/Archie-West HPC, University of Strathclyde, 107 Rottenrow East, Glasgow G4 0NG, UK; karina.kubiak@strath.ac.uk; 3Strathclyde Institute of Pharmacy and Biomedical Sciences, University of Strathclyde, 161 Cathedral Street, Glasgow G4 0RE, UK; v.a.ferro@strath.ac.uk; 4EPSRC Future Manufacturing Research Hub for Continuous Manufacturing and Advanced Crystallisation (CMAC), University of Strathclyde, 99 George Street, Glasgow G1 1RD, UK

**Keywords:** staphylococcus protein A, therapeutics, diagnostics, biomolecular simulation

## Abstract

Staphylococcus protein A (SpA) is found in the cell wall of *Staphylococcus aureus* bacteria. Its ability to bind to the constant Fc regions of antibodies means it is useful for antibody extraction, and further integration with inorganic materials can lead to the development of diagnostics and therapeutics. We have investigated the adsorption of SpA on inorganic surface models such as experimentally relevant negatively charged silica, as well as positively charged and neutral surfaces, by use of fully atomistic molecular dynamics simulations. We have found that SpA, which is itself negatively charged at pH7, is able to adsorb on all our surface models. However, adsorption on charged surfaces is more specific in terms of protein orientation compared to a neutral Au (111) surface, while the protein structure is generally well maintained in all cases. The results indicate that SpA adsorption is optimal on the siloxide-rich silica surface, which is negative at pH7 since this keeps the Fc binding regions free to interact with other species in solution. Due to the dominant role of electrostatics, the results are transferable to other inorganic materials and pave the way for new diagnostic and therapeutic designs where SpA might be used to conjugate antibodies to nanoparticles.

## 1. Introduction

Studying protein adsorption on biomaterials is crucial in modern medicine and biotechnology [1] since it can indicate whether a biomaterial is suitable for a therapeutic application [2]. SpA (Figure 1) is a 42 kDa bacterial protein produced by *Staphylococcus aureus*. It helps the bacteria suppress the host immune response, hence preventing damage to the bacteria [3]. As explained below, the high affinity of SpA for the crystallisable fragment of antibodies makes it an interesting candidate for biotechnologies.

SpA has previously been studied for use as a vaccine candidate, and the effectiveness of this approach relies on its ability to block the immune response [3]. However, such promise has only been observed in mice, and this approach to using SpA in vaccines is still under development. Another therapeutic application for SpA is in Rheumatoid Arthritis (RA), an auto-immune condition where the immune system produces IgG, which attacks the synovial tissues in multiple joints [4], leading to swollen and painful joints and stiffness over a prolonged period of time, exacerbated by resting the joints [5]. An immunoadsorption treatment device containing SpA [6] and ultrapure SpA (PRTX-100) [7] has previously been investigated as a treatment for RA. Furthermore, SpA is also a target for drug delivery; Chen et al. (2019) showed that targeting SpA with monoclonal antibodies significantly decreased *S. aureus* colonisation in the mouse nasopharynx and intestinal tract [8].

An antibody is an immunoglobulin protein that is part of the immune defence system that identifies and neutralises foreign objects [10]. Antibodies are Y-shaped, generally comprising two fragments: the antigen-binding fragment (Fab), which is replicated on each of the upper arms of the Y-shape, and as a highly selective site, binds to antigens. The Fab region is also essential to the adaptive nature of the humoral response and evolves rapidly during an immune response [11] as it changes shape to fit the antigen. The crystallisable fragment (Fc) forms the lower stem [12], and this region links to the Fab region by disulphide bonds, also known as the flexible hinge region [13,14]. The Fc region binds to bacterial proteins such as protein A and protein G [15].

SpA binding to antibodies through the Fc region allows extraction of antibodies while keeping the Fab site free to allow antigen binding [3]. This enables the study of SpA for the development of new diagnostics and therapeutics, where it can be used to conjugate antibodies to nanoparticles. For this concept to succeed, it is crucial to know how the SpA protein interacts with various types of inorganic materials since this will determine whether the SpA maintains its functionality when adsorbed. Molecular Dynamics (MD) simulations can provide molecular-scale insight into the interactions between biomolecules and inorganic materials, revealing details that would otherwise be impossible to observe. The wide availability of simulation software, including bespoke force fields for simulating biomolecular systems, and the availability of high-performance computers has led to the growth of MD as a commonly used simulation technique [16].

The availability of the crystal structure of SpA provides a new opportunity to analyse the promising route of using it to couple antibodies to nanoparticles. Subsequently, it is of crucial importance to comprehensively understand the interactions of SpA with various types of inorganic materials. Ideally, the adsorption process should orientate SpA so that the residues that bind antibody Fc, located between α-helices 7–9, are freely exposed to the solution and that SpA maintains its secondary and tertiary structure to facilitate its functionality. In this paper, the adsorption of SpA to three key types of inorganic surfaces is simulated: (i) hydrophobic, using a model gold surface; (ii) hydrophilic positively charged; and (iii) hydrophilic negatively charged surfaces using model silica surfaces.

## 2. Results

### 2.1. Simulation in Water

To investigate whether the binding regions maintain their structural integrity in the simulation, the SpA structure was placed in a simulation box with water molecules, and the system was neutralised by the addition of NaCl ions at a concentration of 0.02 (mol/L). This system was simulated for 100 ns, and the RMSD is reported in Figure 2a. The RMSD for the whole protein fluctuated between ~3–7 Å throughout the simulation, demonstrating its stability in these simulations (there is no consistent growth in the RMSD and no observable unfolding of the protein on the 100 ns timescale of the simulations). This shows that the model protein structure is suitable enough to be used to assess its adsorption behaviour and interactions with the model surfaces under study. Ramachandran plots, which indicate secondary structural elements, further evidence the protein stability as the protein maintains its secondary structure (Appendix A). The key binding sites of the protein, which bind to the Fc fragment of antibodies, maintain their structural integrity throughout the duration of the simulation. This is crucial for the application of SpA to the binding of antibodies to surfaces and nanoparticles.

The fluctuations observed can mainly be attributed to flexing between the α-helices in the protein, which tend to be linked together by short flexible loops (see Figure 1). The relative stability of the α-helices can be analysed by studying the RMSD for each α-helix separately, overlapping each in turn with its structure at t = 0 ns. The results are also presented in Figure 2a, where it can be seen that α-helix 4 is the most stable in the simulation, with an average RMSD of 0.35 Å, while α-helix 7 is the least stable, reaching a maximum of 2.5 Å at 65 ns and averaging 1.5 Å through the course of the simulation. Consistent bending of α-helix 7 is visible in the simulation. α-helices 8 and 9, the other two with the glutamic acid and aspartic residues involved in binding to the Fc fragment of antibodies, also show suitable stability. It will be particularly interesting to examine how SpA interacts with the model surfaces and explore whether the structural stability remains similar to the stability in the bulk water simulation.

### 2.2. Adsorption on a Negatively Charged Silica Surface

SpA has an overall negative charge of −11e in the simulations, so based on simple electrostatics, there is a net force driving the protein away from the surface since the system is designed to have an electric field above the siloxide-rich surface. However, the surface electric field is screened by the ions in the solution, and the protein experiences a fluctuating electric field when it is closer than the Debye screening length from the surface [17]. Furthermore, the charge distribution across the protein surface is not uniform, meaning a negatively charged protein can approach close to a negatively charged surface, and if it is oriented in a preferred way with positively charged regions exposed to the surface, adsorption is possible [17,18,19].

Adsorption, where the residues show adhesion to the surface and interaction with the ions shielding the surface, is observed with the residues Lys34 and Lys37 (positively charged, strongly hydrophilic), along with Asn36 (neutral, strongly hydrophilic), Met33, and Phe 35 (neutral, hydrophobic), anchoring the protein to the surface, leaving the Fc binding regions, found between α-helices 7–9, freely moving and exposed to the solution (see Figure 3).

Initially, the protein approaches the surface several times without adsorbing, diffusing above the surface due to the effective screening of the surface by the Na^+^ ions (see Figure 3 and Appendix A). During this time, there is no preferential orientation of the protein to encourage anchoring. However, at 77.32 ns in the production trajectory, the Met33 and Lys37 side chains extend and almost make direct contact (come within the relevant VdW radii, defined as the closest distance of two equal non-covalently bound atoms) with the surface. Lys34 is the next residue to adsorb to the surface at 80.24 ns, and the positive charge on the Lys side chains explains the attraction to the negative surface. The Lys34 and Lys37 residues extend their side chains to penetrate the water layer at the ionic surface, displacing the water molecules and creating strong anchors to the surface. The formation of an ordered water layer at ionic surfaces has also been demonstrated previously [17,18,20]. The adsorption in this simulation is electrostatically driven so that the dipole moment of the SpA is oriented towards the O-rich surface. The adsorbed state appears to be stable, and a 100 ns trajectory extension to this simulation (Appendix A) shows the protein remains anchored to the surface in its initial orientation, as dictated by the electrostatics.

The RMSDs reported in Figure 2b show the stability of the protein and the individual α-helices. The protein appears to have similar stability to that in bulk solution (Figure 2a). Again, α-helix 7 is the least stable, with a peak RMSD of 3 Å at 28 ns (see Figure 2b). However, it is clear that lower-index α-helices (so at the N-terminus side) respond to the adsorption processes at around 80 ns, but these are nearest to the silica surface. At the opposite end of the protein, α-helix 8 is the most stable, with a starting RMSD of 0.2 Å, reaching a maximum of 0.9 Å at 88 ns. α-helix 9 also shows suitable stability, although its RMSD fluctuates the most. This is important because it means the end of the protein containing the binding sites is unaffected by the surface adsorption and behaves in a similar way to the protein in bulk solution. Furthermore, the Ramachandran plot shows suitable stability of the protein upon adsorption (Appendix A).

Most importantly, the simulation demonstrates adsorption to the siloxide-rich silica surface, as found experimentally [21], leaving the SpA’s Fc binding region free and exposed to the solution. In this sense, the protein orients itself in an ideal manner due to the role that the electrostatics plays above the charged surface. Despite several attempts, there is only one successful adsorption trajectory in a 100 ns period. Such an adsorption event is rare in a 100 ns timescale because the interacting bodies possess identical charges, and the adsorption requires favourable local fluctuations of the electric field, as well as a bigger system than the other simulations (see Section 4).

### 2.3. Adsorption on a Model Positively Charged Surface

The model silica slab used in the adsorption simulations presents both an oxygen-rich and a silica-rich surface to the solution. The latter creates a model positively charged surface on which SpA adsorption can be explored. The electric field above the surface, while fluctuating due to the movement of the screening Cl^−^ ions, readily pulls the negatively charged protein towards it. We have obtained four independent trajectories for the protein adsorption on this model surface, and a typical one is described here.

The starting orientation of the protein in this simulation is such that the Fc binding sites are not directly facing the surface. SpA rotates in the early stages of simulation so as to align its dipole moment with the electric field above the surface before adsorbing (see Appendix A). The crucial residues involved in the adsorption are Ala221 (neutral, hydrophobic), Ala219 (neutral, hydrophobic), and Gln220 (neutral, strongly hydrophilic), all localised in α-helices 7–9, which is the Fc binding region (Figure 4).

Studying the α-helix stability in this system, α-helix 7 is the most unstable, hitting a peak RMSD of 4 Å at 31 ns. It jumps in value at 30 ns. This is down to the fluctuations of the flexible loops in the structure, linking to a subset of atoms in the helix and causing divergence in the backbone RMSD. In terms of stability, α-helix 8 is very stable, hitting a peak of 0.9 after 45 ns. This is followed closely by α-helix 9, which hits a peak of 1.5 Å at 55 ns (Figure 2c). As it turns out, even the α-helices of interest are generally stable in this simulation, although it is of little consequence given that the residues that bind to the surface are on the antibody binding end of the protein. The Ramachandran plot for this simulation indicates little change in the protein secondary structure throughout the simulation (Appendix A).

### 2.4. Adsorption on a Model Gold Surface

In order to ensure successful binding with the Fc fragment of antibodies, the binding sites found between α-helices 7–9 have to be exposed. The adsorption of the SpA on the Au (111) surface is expected to be non-specific and can be driven by the interactions of several different hydrophobic side chains. This is due to the hydrophobicity of the model gold surface and the lack of long-range electric fields to orient the protein as it approaches the surface. In the adsorption trajectory, one end of the protein (towards the C-terminus) adsorbs first, followed later by the adsorption of the opposite end of the protein (see Appendix A). The final adsorbed conformation is shown in Figure 5. There are five adsorption trajectories on this surface, and a typical one is described here.

The first adsorption event in this simulation occurs at 27.88 ns. This is by Ala221, which is neutral and hydrophobic. This initial interaction on the Au (111) surface occurs at the active binding site of the protein to antibodies, which is evidence of the expected arbitrary adsorption on the Au (111) surface. This is followed by the adsorption of Gln220 (neutral, strongly hydrophilic) at 51.28 ns and Asp218 (neutral, strongly hydrophilic) at 63.08 ns. The adsorption for the other residues on the non-binding side involves Lys34 (positively charged, strongly hydrophobic) at 84.08 ns and Met33 (neutral, hydrophobic) at 88.6 ns. The presence of Met33 in the list of residues interacting with Au (111) is particularly noteworthy as it is a sulphur-containing residue. This is crucial in cysteine, which is the preferential binding site for gold in proteins [22], and dependent on the accessibility, it may be able to play a further role in sulphur–gold interactions, although potential covalent bonding of this type is not considered in these simulations. Note that of these key residues that adsorb to the Au (111) surface, Ala221, Gln220, and Asp218 are located on the Fc binding side of the protein. Although the Ala221 and Gln220 residues are not involved in Fc binding, their interaction with the gold surface impacts those residues that are involved. It is worth noting that the non-specific adsorption on the model hydrophobic surface is likely to encourage the protein to lie parallel to the surface plane so that the availability of the Fc binding residues will always be restricted to some extent. While there is a structured layer of water at the model Au (111) surface, this is penetrated by the SpA residues, which break through the layer to interact directly with the surface; this phenomenon has previously been observed with metal surface models [23,24,25,26].

The RMSD for the protein in the system is relatively stable. All the α-helices in the protein maintain their structural integrity with an average RMSD value of 1.5–2 Å. α-helix 3 goes slightly higher, suggesting more movement for that part of the protein. The most stable α-helix in simulation with the gold surface is α-helix 4, starting at 0.25 Å and consistently maintaining its value throughout, with a top RMSD of 1.3 Å at 12 ns. Although α-helix 4 is not the target α-helix, consistent stability is a suitable sign nonetheless. α-helix 7 shows improved stability in this simulation compared to the water box simulation, starting at 0.7 Å and ending at 1.3 Å, although it is consistently above 1 Å after 13 ns and hits a high of 2.3 Å at 25 ns. α-helix 8 displays instability relative to the simulation in water, hitting a high of 2.1 Å at 67 ns, and at 40 ns, this jumps up significantly. This is an indication of the flexible loops moving together in the active region of the protein. α-helix 9 is slightly more stable, starting at 0.3 Å and finishing at 1.4 Å, with a peak of 1.8 Å at 60 ns (Figure 2d). The Ramachandran plot for this simulation follows the same trend as other simulations, indicating suitable stability and no change in the secondary structure of the protein upon adsorption (Appendix A).

### 2.5. Angles between Neighbouring α-Helices

The protein has nine α-helix structures linked by loop regions, and all consecutive helices are aligned anti-parallel to each other. This is an interesting observation, as visibly, the protein structure moves in unison, and the α-helices stay closely linked throughout, pointing to strong hydrogen bonding, which is optimal in anti-parallel structures. The protein maintains a low RMSD value throughout the simulations, indicating suitable stability and similarity to the original structure.

The angles between the long axes of the neighbouring helices throughout the simulations are given in Figure 6. There is clear uniformity, and the angle is close to (but below) 180° across all simulations, and it is attributed to the nature of the vectors being in opposite directions because of the helix packing. The behaviour of the protein in the bulk water simulation (Figure 6a), where there is no surface to guide protein movement, allows observation of the natural protein behaviour. Adsorption to the model charged surfaces (Figure 6b,c) does not show significantly different fluctuations in the inter-helix angles, again confirming the structural stability of the protein adsorbed to these surfaces.

In the Au (111) simulation, greater fluctuation of some of the inter-helix angles is observed. It is particularly visible in the angle between helices 5 and 6, where the angle is regularly ~13° between 65 and 85 ns. This seems to accommodate the adsorption of both ends of the protein to the surface (see Figure 4); the overall bending of the protein is permitted by hinging movements between the helices, which themselves retain rather rigid structures. In the same simulation, helices 8 and 9 generally maintain a lower angle of ~150° when compared to the bulk water simulation, where it is ~170°.

## 3. Discussion

SpA as a structure has nine fairly rigid α-helices connected by coils and turns. These are responsible for the intrinsic flexibility of the protein, which allows efficient adsorption to the inorganic surfaces. The RMSD values (Table 1) show that SpA maintains the highest structural integrity on the Au (111) surface, and it can be attributed to VdW interactions, which are crucial in hydrophobic interactions [27,28,29]. VdW interactions are when adjacent atoms come close enough so that their outer electron clouds briefly interact. Therefore, they are short-range and relatively weak and are not as specific or strong as electrostatic interactions, though they can still be strong collectively [30]. This explains why SpA maintains its structural integrity on the Au (111) surface; the VdW interactions are strong at a short distance, and the SpA has more time to adjust and find local energetic minima.

Simulations on the two SiO_2_ surfaces show the biggest structural changes for the full protein and for individual α-helices. This links primarily to the electrostatic interactions on the SiO_2_ surfaces, which are strong and long-range [31], so that adsorption is directed and specific. Furthermore, the charge on SpA is not spread evenly across the structure [32], so there are positive and negative patches that can distort the protein’s tertiary structure in the electric field above the charged surfaces [33].

In the adsorption to the negatively charged silica surface simulations, the Na^+^ ions are attracted to the O^−^ siloxide surface, forming a diffuse layer that screens the electric field. Five key residues anchor the protein to the surface, three of which are neutral and two positive. The neutral residues are driven to the surface by the overall charge on the protein, and they do not directly interact with the surface. The positive Lys residues do, however, penetrate the water layers at the surface; this is particularly noticeable with Lys34.

In the adsorption simulation using the Si^+^ under-coordinated surface, the positively charged surface attracts the Cl^−^ ions and creates a diffuse screening. Adsorption in this simulation is driven by three key interacting residues, all of which are neutral. Therefore, the attraction of the protein to the surface is driven by the overall negative charge on the protein, which attracts it to the positive surface below the diffuse Cl^−^ layer. This is supported by three neutral residues that interact with the surface for the duration of the simulation. Due to the orientation of protein as it adsorbs, the Fc binding residues are directed towards the surface and so are not free to interact with the solution. Crucially, it is an important observation because the integration of the SpA in new technology will only really be effective if the Fc binding sites are kept free, allowing interaction of SpA with the antibody, leading to facile and efficient conjugation that enables the antibody to further bind to its target with its exposed Fabs. Consequently, SpA adsorption to positively charged surfaces and nanoparticles is not favourable, while adsorption to negatively charged ones is.

Like many fully atomistic MD simulations, the present work is focused on the first 100 ns or so of protein adsorption on a surface, which is the first stage of adsorption, but not the final stable state, which can be different, as suggested by previous work [34]. Nevertheless, the structural stability of SpA in the simulations was similar to previous work [35]. Furthermore, in the case of the charged surfaces, the electrostatics will tend to keep the protein oriented in the same way as it adsorbs, suggesting that the conclusions from the simulations provide suitable indicators of long-term behaviour, too, as indicated by the 200 ns simulation for the siloxide surface adsorption.

The SiO_2_ slab used to create the model silica surfaces has an intrinsic dipole moment across it; the atoms were fixed in space so that the siloxide (SiO^−^) groups were on the top and under-coordinated Si species at the bottom. When adsorption is driven by a long-range electric field above a positive surface, the adsorption is rapid and specific. In the first stages of the simulation, the dipole moment of the protein quickly aligns (Table 2) with the electric field, and in all simulations, the protein flexes in a way to orient its dipole moment perpendicularly to and away from the under-coordinated surface. In contrast, adsorption to the model hydrophobic gold surface provides no orientational preference for the protein dipole. Since this aligns to the long axis of the protein (see Figure 1), and the protein tends to lie parallel to the surface plane to maximise the short-range interactions with it, the dipole also tends to be parallel to the gold surface (Table 2).

It is worth looking at previous work using similar surfaces and the key residues observed in those simulations. For example, Heinz et al. [36] studied the adsorption of amino acids and surfactants on a model Au (111) surface. Of the key anchoring residues found in the simulations (see Table 3), Gln and Met were found by Heinz et al. to be in the strongest binding group, with Lys having intermediate strength, supporting the stability of the adsorbed state.

Previous work involving protein adsorption on the SiO_2_ surface has revealed roles for many residues. Adsorption of BSA on the SiO_2_ siloxide surface was also reported [17], and it was found that in the final adsorption state, there are several interacting residues at the protein-silica interface, of which Lys plays a significant anchoring role in a similar way to found here (see Table 3). The importance of charged residues in adsorption to the silica surface has been widely noted, in particular the positively charged Lys and Arg residues [37].

## 4. Materials and Methods

The crystal structure of protein A obtained from the Protein Databank entry 5H7A.pdb [38] was used in the simulations. The original structure contained four repeat modules, of which one was used in the simulation. A single SpA unit contains 221 amino acids (3001 atoms), and the net protein charge at pH 7 is −11e. The NAMD 2.12 package [39] was used along with the CHARMM-27 force-field, and the simulation results were analysed with VMD 1.9.1 [9]. The simulation was performed in three stages [17]. The first stage consisted of adding water and ions to the simulation cell that contained the static protein as obtained from the PDB structure, followed by water and ion minimisation (1000 steps) and a subsequent 100 ps run (with integration time-step 1 fs) at a target temperature/pressure of 300 K/1 atm and then in the constant temperature/volume (NVT) ensemble. The second stage performed a complete energy minimisation (10,000 steps) of the whole system (protein + water + ions), followed by equilibration for 300 ps (time-step 1 fs). Finally, the production trajectory was performed, consisting of an initial 10 ns run, extended to 100 ns (with time-step 2fs) at 300 K in the NVT ensemble. Periodic boundary conditions (PBC) were used along with the SHAKE algorithm. The cut-off distance for the van der Waals interactions was 12 Å, and the particle mesh Ewald (PME) summation was used for the Coulomb interactions [40]. Only the production trajectories have been analysed in detail. The simulation of SpA in water (with no surface models) was run for 100 ns for reference. The adsorption simulations were then performed with three different surface models: Au (111) and two model SiO_2_ surfaces with siloxide and under-coordinated silicon terminations, respectively.

Adsorption to the model Au (111) surface was explored by creating a slab of fcc Au oriented with the (111) direction aligned with the z-axis of the simulation cell. The model Au had no partial charges (and so no electric field has been introduced), and the slab dimension was 98 × 196 × 15 Å, with 13,824 Au atoms. The Charmm27 force-field parameters for Au were used following Heinz et al. [36,41,42]. This creates a model hydrophobic surface, mimicking some of the features of a gold surface; however, no polarisation forces are included in this model, and chemical bonding with sulphur was not considered.

The model SiO_2_ slab was also neutral, but there was an intrinsic dipole moment across it that did not change during the simulation; the slab was created from an α-cristobalite structure, and the material was modelled as ions fixed in space [21]. The 3D periodicity of the simulation cell creates a compensating electric field across the protein/water space in response to the fixed dipole moment of the slab [43] so that the electrostatic environment above the slab mimics that expected above a charged surface. The SiO_2_ slab dimension was 103 × 199 × 13 Å, with 17,280 atoms, and has two different surfaces that are normal to the simulation cell z-axis. The SiO_2_ surface with siloxide (SiO^−^) groups at the top mimics the negatively charged surface encountered experimentally at pH7 [21]. The surface with under-coordinated Si species exposed mimics a positively charged surface; this is not encountered in practice, except perhaps at very low pH, but nevertheless provides an interesting model structure to explore SpA adsorption at a hypothetical positively charged material surface. The Charmm27 force-field parameters for the surface were used according to Patwardhan et al. [44]. The electric field that runs in the z-direction between the top of the SiO_2_ slab to the bottom of its image (across the water-protein space) is screened by NaCl: 288 Na^+^ ions to screen the siloxide-rich surface and 277 Cl^−^ ions to screen the positive Si-rich surface. The additional 11 Na^+^ ions neutralise the system, as the protein has a charge of −11 at pH 7.

The adsorption simulations were prepared on the Au (111) surface by placing the protein ~20 Å above the surface, and the system was solvated in a water box extending at least 30 Å above the protein, resulting in a system with ~160,000 atoms. The system for adsorption on the Si-rich SiO_2_ surface involved placing the protein ~40 Å above the surface, and the system was solvated in a water box extending 60 Å above it. This provided a system with ~260,000 atoms. The system for adsorption on the siloxide-rich surface had the protein placed ~80 Å above the surface, and a water box of a further 40 Å, giving a system with ~310,000 atoms. This larger system was employed to allow the protein time to diffuse and re-orientate before adsorbing to the surface.

## 5. Conclusions

In the present study, fully atomistic MD simulations of SpA at different model surfaces are presented. A model SiO_2_ slab was used with an electric field across the water/protein space and a model Au (111) slab without an electric field. When the adsorption was driven by a long-range electric field above a positive surface, as seen with the under-coordinated silica surface model, the adsorption was found to be direct and specific. The dipole moment of the protein quickly aligns with the electric field in the early stages of the simulation, and the anchoring residues interact specifically with the surface. Similarly, adsorption to the negatively charged siloxide surface required the protein to re-orientate above the surface, which it is able to do due to the screening of the electric field by the diffuse layer of counter ions in solution. SpA has a large negative charge (−11e) at pH7 yet can still adsorb to the negatively charged silica surface due to the distribution of positively and negatively charged residues on the protein.

The adsorption on the uncharged Au (111) surface was relatively slow and non-specific, and the anchoring residues involved were mostly uncharged. The interactions are, therefore, short-range in nature, with no preferred protein orientation other than to maximise the contacts between the protein and the surface atoms. For this reason, the long axis of the protein tends to lie parallel to the surface plane, which therefore prevents SpA’s Fc binding residues from freely interacting with the solution. This low adsorption specificity on gold has also been found by other authors [36,41,42], and since hydrophobic Met33 is involved in the protein-gold interactions, this might indicate an important role in sulphur–gold interactions.

The conformational changes of SpA observed upon adsorption were related to local structural adjustments facilitated by the intrinsic flexibility between the α-helix structural elements of the protein. The Ramachandran plots (see Appendix A) highlight SpA as a helical protein, and no protein unfolding is observed. It is to be noted that only the early stages of protein adsorption were assessed, so the long-term adsorption states might be different. This is particularly true for the adsorption onto the uncharged hydrophilic model gold surface. However, for the adsorption onto the charge silica surfaces, it is worth noting that the electrostatics favours the adsorbed orientation of the protein with its long axis directed away from the surface plane so that this is likely to be the long-term orientation.

Taken together, these results indicate that adsorption to the negatively charged silica surface, as observed experimentally for silica nanoparticles, is likely to produce favourable SpA adsorption that facilitates the binding of antibodies at the Fc region to functionalise the system. These results are, to some extent, transferable to other inorganic materials, and adsorption to other negatively charged nanoparticles would similarly result in favourable functionalisation. The results presented here, therefore, pave the way to creating versatile methods to create new diagnostics and therapeutics.

## Figures and Tables

**Figure 1 ijms-23-04832-f001:**
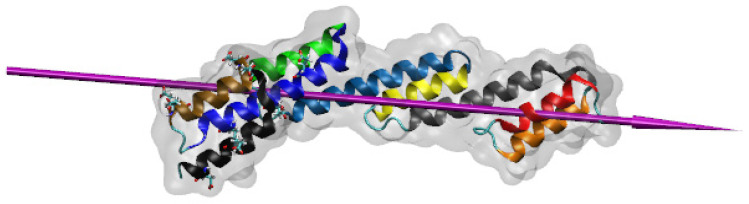
The SpA structure (PDB: 5H7A) illustrated using VMD 1.9.1 [9]. The protein is indicated as a ghost surface, and the secondary structural α-helices are coloured as follows. Helix 1: red ribbons (close to the N-terminus), helix 2: orange, helix 3: grey, helix 4: yellow, helix 5: light blue, helix 6: green, helix 7: dark blue, helix 8: brown, helix 9: black (close to the C-terminus). The purple needle indicates the dipole moment. The key Glu and Asp residues that interact with the Fc fragment of antibodies are indicated in the ‘liquorice’ representation [9].

**Figure 2 ijms-23-04832-f002:**
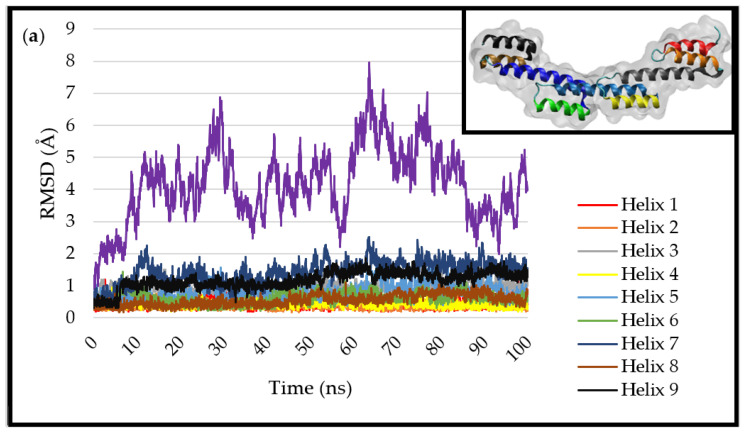
RMSD of SpA in (**a**) water, (**b**) siloxide surface, (**c**) under-coordinated surface, (**d**) Au (111) surface simulations. The RMSD for the individual α-helices follows the colour scheme in Figure 1. The purple line represents the RMSD for the full protein, including the flexing between the α-helices.

**Figure 3 ijms-23-04832-f003:**
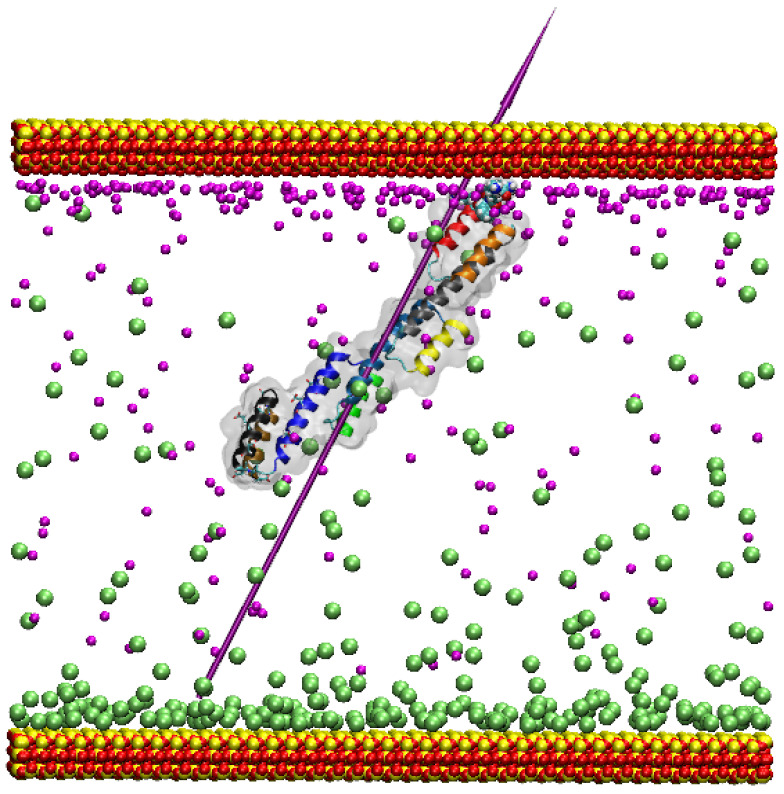
Adsorption of SpA on the oxygen-rich surface. The protein colour scheme follows that in Figure 1. The outer layer of oxygen atoms in the silica model surface is shown as red spheres. Cl^−^ ions are shown as lime VdW spheres and Na^+^ as magenta. The adsorbing residues are also shown as VdW representation. The purple needle indicates the dipole moment, and water molecules are not shown for clarity.

**Figure 4 ijms-23-04832-f004:**
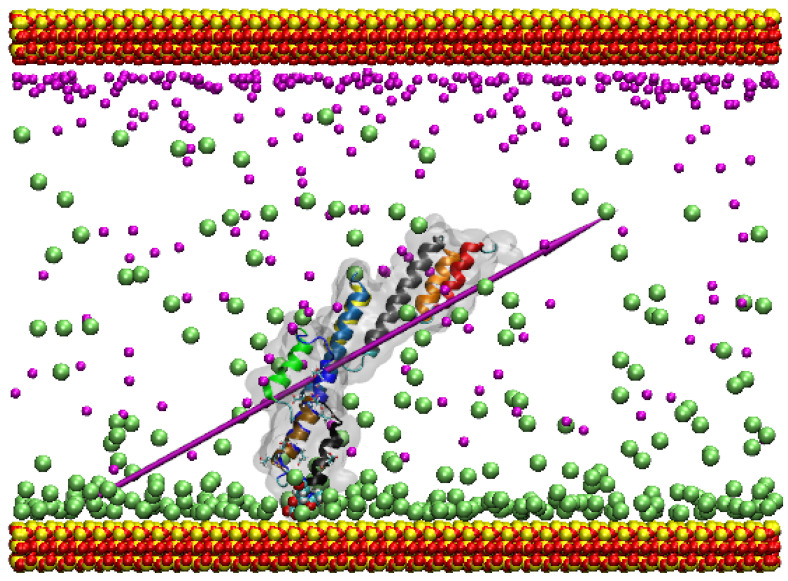
Adsorption of SpA on the positively charged surface. The colour scheme is the same as in Figure 3. The purple needle indicates the dipole moment, and water molecules are not shown for clarity.

**Figure 5 ijms-23-04832-f005:**
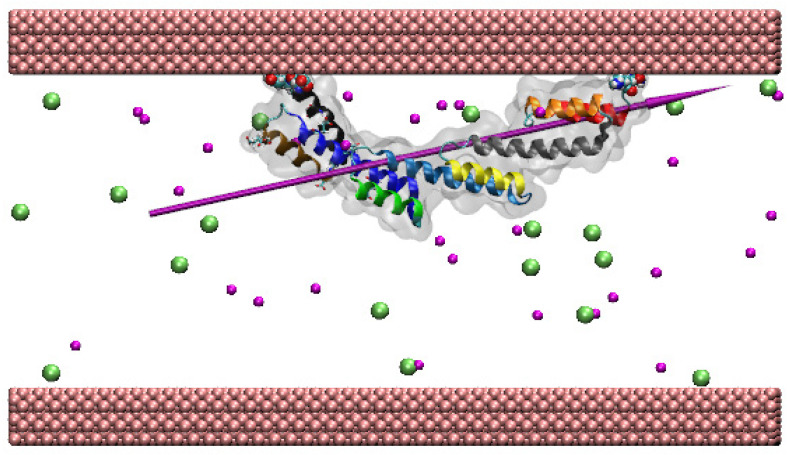
Adsorption of SpA on the hydrophobic gold surface. The colour scheme is the same as in Figure 3. The gold atoms are shown as pink VdW spheres (defined in VMD 1.9.4). The purple needle indicates the dipole moment, and water molecules are not shown for clarity.

**Figure 6 ijms-23-04832-f006:**
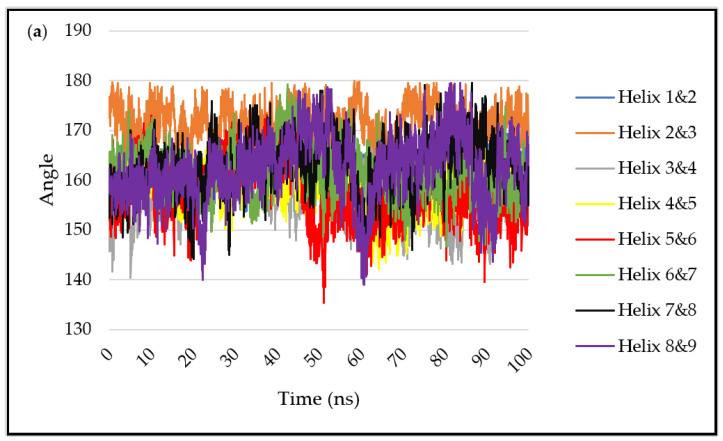
The angles between the α-helix pairs (**a**) water, (**b**) siloxide, (**c**) Si under-coordinated, (**d**) Au (111).

**Table 1 ijms-23-04832-t001:** RMSD summary in water, Au (111), SiO_2_ O^−^, SiO_2_ Si^+^ simulations.

	RMSD Values in Simulation											
	Start (Å)–0 ns				End (Å)–100 ns				Max RMSD Value (Å)			
Simulation	Water	Au (111)	SiO_2_ O^−^	SiO_2_ Si^+^	Water	Au (111)	SiO_2_ O^−^	SiO_2_ Si^+^	Water	Au (111)	SiO_2_ O^−^	SiO_2_ Si^+^
**α-helix1**	0.2	0.2	0.29	0.2	1	0.3	0.2	0.2	1.1	1	0.88	1.4
**α-helix2**	0.32	0.3	0.23	0.4	1	0.33	0.2	0.6	0.95	1.3	1.07	1.2
**α-helix3**	0.33	0.35	0.6	0.7	1	1.35	1	1.2	1.4	2.7	1.7	1.9
**α-helix4**	0.3	0.25	0.3	0.3	0.35	0.25	1	0.5	0.75	1.3	1.1	1.4
**α-helix5**	0.5	0.4	0.6	0.5	0.8	1.3	1.4	1.5	1.6	1.48	2.2	1.8
**α-helix6**	0.3	0.35	0.3	0.4	0.6	0.5	0.6	0.4	1.4	1.1	1.2	1.2
**α-helix7**	0.5	0.7	0.6	0.6	1.7	1.3	1.5	3.6	2.5	2.3	3	4
**α-helix8**	0.3	0.3	0.2	0.2	0.7	1.5	0.2	0.5	1	2.1	0.9	0.9
**α-helix9**	0.5	0.3	0.3	0.5	1.3	1.4	0.3	1.1	2.1	1.8	1.4	1.5
**Full Protein**	1	1	3	2	4	5.8	6.3	5.1	7.8	6.5	8	8

**Table 2 ijms-23-04832-t002:** SpA dipole moment at various stages of adsorption on the three different surfaces. 0 = starting position, D1/D2 = protein movements in solution, pre-adsorption, abs = at the time of adsorption, 10/20 = 10/20 ns post adsorption. The arrows indicate the direction of the dipole moment vector. The upwards direction is normal to the surface for the Si-rich surface so that the surface normal points in the downwards direction for the O-rich surface. Numbers in the following row indicate the value of the dipole in Debyes.

	Simulation Time and Dipole (Debyes)
Surface	0	D1	D2	abs	10	20
Au (111)	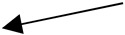	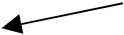	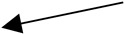	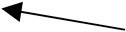	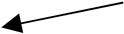	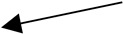
	883	941	928	850	980	973
Silica siloxide-rich	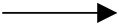	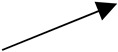	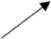	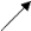	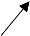	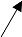
	883	971	902	1066	1036	993
Silica under-coordinated Si	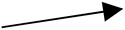	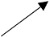	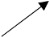	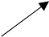	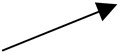	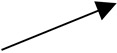
	882	947	942	1055	1027	1175

**Table 3 ijms-23-04832-t003:** Residues in contact with the surface (anchoring residues) in the final stages of SpA adsorption on the three model surfaces.

Surface	Adsorbing Residues
Au (111)	Ala221, Gln220, Asp218, Met33, Lys34
Silica O-rich	Met33, Lys34, Phe35, Asn36, Lys37
Silica Si-rich	Ala221, Gln220, Ala219

## Data Availability

The data presented in this study will be made available in the University of Strathclyde PURE data repository (https://doi.org/10.15129/1573db95-b06c-4dfd-b54b-eb0041f537e1 (accessed on 30 March 2022)).

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
