# Peer review of "Functionalisation of Inorganic Material Surfaces with Staphylococcus Protein A: A Molecular Dynamics Study"

_ijms, 2022, doi:10.3390/ijms23094832_

Round 1
Reviewer 1 Report
In this paper is reported the study of the adsorption of the Staphylococcus protein A with different inorganic surfaces using MD simulations.
The methods are adequately described and conclusions are supported by data. I have only minor concerns about the results:
- it is known that Au (and other metallic surfaces) creates over them a polarized layer of water that is dependent on the Miller indexes of the surface and that can inhibit the adsorption of the proteins. See for example https://doi.org/10.1021/acs.jpcb.7b07492. Did authors noticed something similar for the studied protein?
- Did authors noticed the same trend in the formation of the water layer over the Si- surfaces?
- Figure 1 is in low resolution and I suggest to authors to include in the figure the labels indicating the N-terminal and C-terminal so the reader can better understand the protein orientation
- The plot of RMSD are interesting, but in the figure of the protein there is no purple helix (or I cannot see it). Can authors label this helix or change the colors?
- It seems that the protein does not have appreciable structural changes. Did authors computed the Ramachandran plot or the percentage of secondary structures before and after the adsorption?
Author Response
21st April 2022
Dear Ms. Frieda Kang,
Thank you for sending the reviewers’ feedback regarding our submitted article, and the helpful suggestions to improve it. Below we respond to each point in turn, and give details of the changes we have made to the revised manuscript.
Yours sincerely,
Mohammed Farouq, Karina Kubiak-Ossowska, Mohammed M. Al-Qaraghuli, Valerie Ferro and Paul Mulheran
Reviewer 1:
Comment 1: it is known that Au (and other metallic surfaces) creates over them a polarized layer of water that is dependent on the Miller indexes of the surface and that can inhibit the adsorption of the proteins. See for example https://doi.org/10.1021/acs.jpcb.7b07492. Did authors noticed something similar for the studied protein?
Response: This is a good point, and yes this layering is present in our simulations. We have added a brief discussion on lines 238-241 on page 8 of the revised manuscript.
Comment 2: Did authors noticed the same trend in the formation of the water layer over the Si- surfaces?
Response: Yes again; this point has been addressed by adding more information on lines 149-154 on page 6 of the revised manuscript.
Comment 3: Figure 1 is in low resolution and I suggest to authors to include in the figure the labels indicating the N-terminal and C-terminal so the reader can better understand the protein orientation
Response: We have added an indication of the N and C terminus to the Figure 1 caption on lines 52-54 on page 2; we feel this is sufficient and avoids cluttering the figure with too many labels.
Comment 4: The plot of RMSD are interesting, but in the figure of the protein there is no purple helix (or I cannot see it). Can authors label this helix or change the colors?
Response: On the RMSD figures, the purple line indicates the full protein which includes the flexing between the α-helices. This has been added to the Figure 2 caption, lines 119 and 120 on page 5.
Comment 5: It seems that the protein does not have appreciable structural changes. Did authors computed the Ramachandran plot or the percentage of secondary structures before and after the adsorption?
Response: The Ramachandran Plots have been computed before and after simulation, and are available as supplementary material. Additional information has also been added to lines 96-98 on page 3, lines 169 and 170 on page 6, lines 204-206 on page 7, lines 254-256 on page 9 and lines 448, 449, 464 and 465 on page 15 of the revised manuscript.

Reviewer 2 Report
In this work, the authors investigates adsorption of Spa protein on negatively charged silica surface and on a model positively charged surface. The work can potentially show meaning insights on the protein adsorption on these surfaces. While the manuscript is nicely presented and well organized, the main concern I have is that 100 ns is very short to support the authors claim on the protein adsorption. Indeed on Page 12, the authors mentions "the final stable state which can be different" and so the simulation must be repeated for a longer time duration or provide a more stronger reasoning to show that 100 ns is enough. Furthermore, this statement, "the electrostatics will tend to keep the protein oriented in the same way as it adsorbs" (on page 12) needs an elaborate explanation and argument for such claims. I recommend to consider the manuscript for publication only after addressing this main concern.
Author Response
21st April 2022
Dear Ms. Frieda Kang,
Thank you for sending the reviewers’ feedback regarding our submitted article, and the helpful suggestions to improve it. Below we respond to each point in turn, and give details of the changes we have made to the revised manuscript.
Yours sincerely,
Mohammed Farouq, Karina Kubiak-Ossowska, Mohammed M. Al-Qaraghuli, Valerie Ferro and Paul Mulheran
Reviewer 2:
Comment 1: In this work, the authors investigates adsorption of Spa protein on negatively charged silica surface and on a model positively charged surface. The work can potentially show meaning insights on the protein adsorption on these surfaces. While the manuscript is nicely presented and well organized, the main concern I have is that 100 ns is very short to support the authors claim on the protein adsorption. Indeed on Page 12, the authors mentions "the final stable state which can be different" and so the simulation must be repeated for a longer time duration or provide a more stronger reasoning to show that 100 ns is enough. Furthermore, this statement, "the electrostatics will tend to keep the protein oriented in the same way as it adsorbs" (on page 12) needs an elaborate explanation and argument for such claims. I recommend to consider the manuscript for publication only after addressing this main concern.
Response: As mentioned on lines 156-159 on page 6 and lines 331 and 332 on page 12 of the revised manuscript, one of the simulations was extended for an additional 100ns and the protein showed good adsorption stability for 100-200ns. This supports the contention that atomistic simulations can indicate longer term behaviour, however it is important that we acknowledge this limitation of the methodology which cannot be overcome by longer simulation trajectories. We have also added a longer simulation trajectory to the Supplementary Information (“Siloxide200ns.mp4”), this is indicated on line 464 on page 15.
